# PACER: PHYSICS INFORMED AND UNCERTAINTY AWARE CLIMATE EMULATOR

## ABSTRACT

Physics based numerical climate models serve as critical tools for evaluating the effects of climate change and projecting future climate scenarios. However, the reliance on numerical simulations of physical equations renders them computationally intensive and inefficient. While deep learning methodologies have made significant progress in weather forecasting, they are still unstable for longer roll-out climate emulation task. Here, we propose **PACER**, a relatively lightweight 2.1M parameter **P**hysics Informed Uncertainty **A**ware **C**limate **E**mulato**R**. PACER is trained across is trained across varying spatial resolutions and physics based climate models, enabling faithful and stable emulation of temperature fields at multiple surface levels over a 10 year horizon. We propose an auto-regressive ODE–SDE framework for climate emulation that integrates the fundamental physical law of advection, while being trained under a negative log-likelihood objective to enable principled uncertainty quantification of stochastic variability. We show PACER's emulation performance across 20 climate models outperforming relevant baselines and advancing towards explicit physics infusion in ML emulator. The code is available at `https://anonymous.4open.science/r/PACER-B7C3/`.

## 1 INTRODUCTION

Data-driven weather forecasting models Kochkov et al. (2024); Lam et al. (2023); Nguyen et al. (2023b) have outperformed Numerical Weather Prediction (NWP) (ECMWF, 2023) in recent years for short- to medium-range targets, but their long-lead auto-regressive roll-out becomes numerically unstable over multi-year horizons, undermining reliability for climate modeling (Chattopadhyay & Hassanzadeh, 2023). This necessitates the use of dynamical climate models known as general circulation models (GCMs), which evolve the state forward under specified forcings to yield physically consistent decadal projections.

Physics based dynamical climate models are governed by temporal partial differential equations (PDEs) to describe complex physical processes Gupta & Brandstetter (2022), enabling simulations of climate behavior under various forcing scenarios, such as fluctuating greenhouse gas (GHG) emissions. However, the computational expense associated with solving these PDEs involves the execution of these climate model simulations typically for several months (Balaji et al., 2017).

With the advancement of Artificial Intelligence (AI), data-driven emulators provide a computationally viable substitute. In order to faithfully emulate the dynamical physics based climate model, a Machine Learning (ML) based climate emulator must respect the fundamental physical laws that govern the dynamics of the atmosphere (Watt-Meyer et al., 2023) which improves reliability. Furthermore, accurately capturing the influence of GHG and aerosols is essential for simulating realistic climate responses to different emission scenarios (Bloch-Johnson et al., 2024).

Recently, climate emulators such as ACE (Ai2 Climate Emulator) (Watt-Meyer et al., 2023) and Spherical Dyffusion (Cachay et al., 2024) have shown remarkable and stable results for over 10 year emulation. ACE exhibits physically consistent emulations under prescribed, time-varying sea surface temperatures (SST) and sea-ice boundary conditions, but its deterministic formulation limits the representation of the predictive distribution of climate states or support ensemble generation which aids in better uncertainty quantification. Spherical Dyffusion solves this problem by training on dynamics informed diffusion model known as Dyffusion Cachay et al. (2023) but do not take into

account the projected GHG emissions which are essential to the faithful emulation of the climate. Furthermore, both architectures are parameter-heavy, with 200M parameters.

To address these gaps, we propose PACER, a lightweight, physics informed, Neural stochastic differential equation (sde) based, 2.1M parameters climate emulator. PACER integrates GHG emissions data directly into the model's training framework, to predict climate variables from a given time-series of climate forcer emission maps (GHG and aerosols) allowing for more accurate simulation of future climate states under varying concentration scenarios. Furthermore, we focus on a key phenomenon driving our climate system i.e. advection. In climate modeling, the advection equation is fundamental for simulating the transport of climate variables, such as temperature and moisture (Choi et al., 2023).

Furthermore, PACER is trained with multiple stochastic realizations under direct Negative-log likelihood (NLL) objective which accounts for aleatoric uncertainty and reduces gradient variance. Additionally, modeling the key physical law of advection reduces its dependence on large datasets making it data-efficient and more generalizable across different climate models. Our contributions are as follows:

1. **Physics-guided backbone:** We propose PACER with a physics guided backbone and integrate a deterministic advection-forcing PDE by dynamically estimating flow velocities to generate a physically consistent trajectory.

2. **Stochastic residual corrector:** We refine the ODE trajectory with a stochastic residual corrector. A neural SDE, that learns unresolved/process noise and systematic bias, improving long-horizon fidelity without sacrificing physical structure.

3. **Likelihood-based uncertainty:** We train the Neural SDE on NLL objective with multiple Brownian realizations per sample ($K > 1$), yielding explicit, heteroscedastic aleatoric uncertainty and reducing gradient variance $\propto 1/K$ for more stable optimization.

4. Finally, we perform extensive experiments across 20 physics based climate models to show PACER is physically consistent and stable for 10 year auto-regressive roll-outs. We also perform zero-shot emulation to highlight generalization capabilities of PACER.

## 2 RELATED WORK

**Machine Learning (ML) and Physics based Climate Emulators:** Physics based dynamical climate models underpin assessments of future change. These models solve discretized PDEs on a global grid and are computationally prohibitive for multi-decadal integrations, limiting their direct use. Recently, hybrid ML-based and physics-informed climate emulators have been successful in emulating several climate variables at lower wall-clock cost, with strong fidelity across intermediate climates. In literature, climate emulation has been treated as either auto-regressive or diagnostic task. Spherical Fourier Neural Operator (SFNO) based auto-regressive models have shown impressive emulation skills. Cachay et al. (2024) proposed Spherical Dyffusion, a generative emulator of the coarse-grid dataset FV3GFS (see Section 5), which couples dynamics-informed diffusion (DYffusion) with a SFNO.Watt-Meyer et al. (2023) proposed ACE (AI2 Climate Emulator) also based on SFNO for effective physics informed emulation. Another version of ACE (ACE2-ERA5) (Van Loon et al., 2025) is trained on reanalysis data to observe radiative response to changing sea surface temperature patterns. LUCIE and LUCIE3D (Guan et al., 2024; 2025) are also trained on reanalysis data to account for the computational complexity of ACE.

Choi et al. (2023) proposed climate modeling using Graph Neural Networks (GNNs) and Neural ODEs, but do not account for GHG emissions or show any long term stability. Additionally, there are several climate emulators accounting for emissions and either treat emulation as diagnostic emulation task (Kaltenborn et al., 2023) or single-model emulators that replicate a specific GCM (Scher, 2018; Mansfield et al., 2020; Beusch et al., 2020; Cachay et al., 2021; Watson-Parris et al., 2022; Nguyen et al., 2023a; Kaltenborn et al., 2023).

## 3 BACKGROUND

**Neural SDE:** A stochastic differential equation (SDE) models continuous-time evolution with random forcing:

$$\mathrm{d}x_t = f(x_t, t)\,\mathrm{d}t + g(x_t, t)\,\mathrm{d}W_t \tag{1}$$

where $f$ is the drift, $g$ the diffusion, and $W_t$ a Wiener process better known as Brownian motion. The Brownian term injects state/time-dependent stochastic forcing via $g(x_t, t)$. In Ito form, the state density $p(x, t)$ obeys the forward Kolmogorov (Fokker–Planck) equation describing the time-evolution of the PDE and the related Stratonovich form corresponding the classical chain rule and is coordinate-invariant, which is often advantageous in physics.

Neural SDE introduced by Kidger et al. (2021) replaces $f$ and $g$ with neural networks $f_\theta$ and $g_\theta$ (Shen & Cheng, 2025). The result is a learned stochastic flow that is continuous in time and probabilistic, which naturally handles irregular sampling, missing data, variable horizons and encodes heteroscedastic aleatoric uncertainty. Training a neural SDE on negative log-likelihood objective with re-parameterized Brownian paths gives calibrated and input-dependent uncertainty.

Wiener (Brownian) process is a continuous path with independent Gaussian increments, so over a step $\Delta t$, the increment satisfies $\Delta W_t \sim \mathcal{N}(0, \Delta t)$, where $\mathcal{N}$ is normal (Gaussian) distribution with mean 0 and variance $\Delta t$. In the SDE, $\mathrm{d}W_t$ provides white-noise forcing representing fast, unresolved processes. In discrete time the increment is sampled as $\Delta W_t \approx \sqrt{\Delta t}\,\varepsilon$ with $\varepsilon \sim \mathcal{N}(0, I)$ with standard multivariate normal and identity covariance $I$. In equation 1, the drift network $f_\theta$ sets the mean tendency, while the diffusion $g_\phi(x, t)$ scales this stochastic input state-dependently.

Sampling different Brownian sequences $\{\varepsilon_k\}$ yields different trajectories from the same initial condition, forming an ensemble, aggregating many such realizations characterizes the predictive distribution and when used in NLL training, reduces gradient variance by averaging over noise paths.

## 4 METHODOLOGY

### 4.1 PROBLEM STATEMENT

We model climate emulation as a probabilistic continuous-time, auto-regressive task $P_\theta\big(x_{t_1:t_H} \,\big|\, x_{t_0}, f_{[t_0, t_H]}\big)$. The goal is to learn the climate variable trajectory $(x_{t_1:t_H})$ over a time horizon $H$ given an initial state $x_{t_0} = x_0 \in \mathbb{R}^{C \times X \times Y}$, where $C$ is the 2D climate variable of interest driven by time-varying GHG forcings $F$, with $f_t \in \mathbb{R}^{F \times X \times Y}$.

The dynamics are learned over two stages.

**Physics backbone ODE:** We first integrate an advection-forcing partial differential equation (PDE) to obtain a deterministic physics trajectory, considering that climate system evolves according to a 2D advection-forcing process.

**Neural SDE refinement:** We then refine the ODE physics trajectory with a neural SDE residual driven by multiple Brownian paths to capture unresolved variability.

Formally, let $x_t \in \mathbb{R}^{C \times X \times Y}$ be the climate state at time $t$ on $X \times Y$ latitude-longitude spatial grid and $F_{t:t+\Delta t} \in \mathbb{R}^{F \times X \times Y}$ be prescribed GHG forcings over $[t + \Delta t]$. We learn the climate trajectory as

$$x_{t+\Delta t} = \mathcal{M}_\theta\big(x_t, F_{t:t+\Delta t}\big) = \underbrace{\Phi_{\Delta t}\big(x_t, F_{t:t+\Delta t}\big)}_{\text{ODE: advection + forcings}} + \underbrace{R_\theta\big(x_t, F_{t:t+\Delta t}\big)}_{\text{neural residual refinement}} \tag{2}$$

where $\mathcal{M}_\theta$ is the learned one-step update operator that maps the current state and forcings to the next state. $\Phi_{\Delta t}$ is the ODE solver and $R_\theta$ is the neural residual module parameterized by $\theta$, which refines the solution to the advection-forcing equation given the input emissions $F$. Iterating this step auto-regressively from $x_0$ under $F_{0:H}$ yields the trajectory to the time horizon $H$. The model is thus designed to learn the spatiotemporal patterns of our climate system dictated by the underlying physical processes modeled by the PDE.

## 4.2 ADVECTION PROCESS

We model climate emulation as a continuous spatio-temporal process that captures the fundamental physical process of advection under GHG forcings, which together dictate how substances are transported throughout the climate system. The general form of the advection-forcing equation in a climate system is defined in equation 3.

$$\frac{\partial u}{\partial t} + v_x \frac{\partial u}{\partial x} + v_y \frac{\partial u}{\partial y} = S(u, F; x, y, t),\tag{3}$$

where $u$ is the physical variable under consideration and $S$ represents the ghg forces. To faithfully emulate the climate's chaotic nature, it is essential to determine the path and rate at which the physical quantities are transported given by $v \cdot \nabla u$ where $v$ is the velocity vector of the fluid (e.g., wind velocity) and $\nabla u$ is the gradient of the quantity being transported (e.g., temperature).

We use an ODE solver to solve the 2D advection-forcing equation 3 by discretizing the spatial domain using the centered Finite Difference Method (FDM) Fiadeiro & Veronis (1977) considering the earth is divided into spatial grid points in x and y directions (longitude x latitude). FDM employ spatial discretization to approximate derivatives using the values at grid points (Molenkamp, 1968; LeVeque, 2007). We explain the spatial discretization and show it's effect visually in Appendix B.3. The spatial derivatives are therefore descritized as equation 4

$$\frac{\partial u}{\partial x}(x_i, y_j, t^n) \approx \frac{u_{i+1,j}^n - u_{i-1,j}^n}{2\,\Delta x}, \qquad \frac{\partial u}{\partial y}(x_i, y_j, t^n) \approx \frac{u_{i,j+1}^n - u_{i,j-1}^n}{2\,\Delta y}.\tag{4}$$

The resulting semi-discrete system (continuous in $t$) is integrated with a Dormand–Prince ODE solver (dopri5) (Dormand & Prince, 1980).

## 4.3 VELOCITY INFERENCE

Quantifying the temporal change in velocity of climate variables over the sampling grid is essential for accurate climate modeling. Because global and regional models evolve climate fields on a fixed grid, their temporal trends must be translated into spatial displacements. The standard approach is the climate-velocity index (Gaponenko et al., 2022). For a state variable $u$ (e.g., temperature), the local migration speed is given by equation 5

$$v(x) = \|\nabla u\|\, \frac{\partial u}{\partial t}\tag{5}$$

Here, we infer a vector transport field $\mathbf{v}(x, y, t)$ by inverting an advection balance for the predicted state variables. We empirically estimate the initial velocity from data using temporal derivative from a natural cubic spline fit McKinley & Levine (1998), and spatial gradients via finite differences, then improve the initialization with a small residual multi-layer perceptron during ODE integration to obtain a directional flow field used prognostically in the advection–forcing ODE.

**Velocity initialization from past states.** Let $u_{\text{past}} \in \mathbb{R}^{B \times T \times C \times \text{lat} \times \text{lon}}$ be a short history (where $T = timesteps$) of $C$ *state/target channels* on a longitude-latitude grid. For each channel $i \in \{1, \ldots, C\}$ and grid point $(x, y)$, fit a natural cubic spline $s_i(t; x, y)$ to $\{u_{\text{past}}(\cdot, t_k, i, x, y)\}_{k=0}^{T-1}$ and extract

$$\dot{u}_i(x, y) = \partial_t s_i(t; x, y)\big|_{t=t_0}.$$

We compute centered finite-difference spatial gradients at $t_0$,

$$\partial_x u_i(x, y), \qquad \partial_y u_i(x, y),$$

and a stabilizing Laplacian

$$\Delta u_i(x, y) = \partial_{xx} u_i(x, y) + \partial_{yy} u_i(x, y).$$

We set $\alpha = 10^{-7}$; this is a small stabilization constant to avoid division by near-zero $\partial_x u, \partial_y u$. as shown in equation equation 6

$$v_x \mathrel{+}= \frac{-\dot{u}_i - u_i\,\Delta u_i}{\partial_x u_i + \alpha}, \qquad v_y \mathrel{+}= \frac{-\dot{u}_i - u_i\,\Delta u_i}{\partial_y u_i + \alpha}\tag{6}$$

**Velocity refinement:** Given the current state $u(t) \in \mathbb{R}^{B \times C \times \text{lat} \times \text{lon}}$ and previous velocity $v(t) \in \mathbb{R}^{B \times C \times \text{lat} \times \text{lon}}$, we implement a simple 2-layer $1 \times 1$ convolutional MLP which produces

$$\Delta v(u) \;=\; \underbrace{W_2^{(1\times1)}}_{32\to2} \; \sigma\left(\underbrace{W_1^{(1\times1)} * u}_{2\to32}\right). \tag{7}$$

where $*$ is a convolution operator, $\sigma$ represents a ReLU non-linearity, $W_n$ terms are the layer $n$ weights, and applies a residual update

$$v_{\text{ref}}(t) \;=\; v(t) \;+\; \Delta v\big(u(t)\big).$$

The advection–forcing dynamics use $v_{\text{ref}}$,

$$\partial_t u \;=\; -\,v_{\text{ref},x}\,\partial_x u \;-\; v_{\text{ref},y}\,\partial_y u \;+\; S(u, F),$$

and the velocity relaxes toward the refined field,

$$\partial_t v \;=\; v_{\text{ref}} \;-\; v,$$

The refined velocity is then used in the original pde solved by dopri5 solver in equation 3. The refiner learns a state-dependent residual and learns transport corrections directly from loss signals on $u$, improving downstream forecast error without breaking the physics structure.

### 4.4 NEURAL SDE FOR TRAJECTORY REFINEMENT

The physics-based trajectory produced by the ODE is then refined using a unet based Neural SDE. Given an ODE–advanced state sequence $x_{\text{next}}(t)$ over a short window, we encode it to a latent initial condition given in equation 8

$$z_0 \;=\; \mathcal{E}\big(x_{\text{next}}\big). \tag{8}$$

where $\mathcal{E}$ is a 3D encoder that maps the physical fields in the current window to a compact latent state $z_0$ that summarizes the flow-relevant information for stochastic refinement. We then evolve $z(t)$ by an Itô SDE with *diagonal* noise given by equation 9

$$\mathrm{d}z_t \;=\; f_\theta(z_t, t)\,\mathrm{d}t \;+\; g_\theta(z_t, t)\,\mathrm{d}W_t, \qquad z_{t_0} = z_0, \tag{9}$$

where $W_t$ is a $d$–dimensional Wiener process, $f_\theta$ (drift) and $g_\theta$ (diffusion) are convolutional maps defined on the latent grid. We parameterize the SDE coefficients as *convolutional operators* on a latent grid. Let $\mathcal{Z}$ denote the latent field space (channels$\times$height$\times$width). We learn

$$f_\theta : \; \mathcal{Z} \to \mathcal{Z}, \qquad g_\theta : \; \mathcal{Z} \to \mathcal{Z}, \tag{10}$$

where $f_\theta$ (drift) is implemented as a *multiscale unet style encoder-decoder with skip connections* followed by a $1 \times 1$ convolution to produce a residual tendency, and $g_\theta$ (diffusion) is a shallow 2D convolutional encoder–decoder that maps the latent state $z_t$ to a per-pixel, per-channel diffusion amplitude for the Neural SDE. This construction preserves spatial locality and translation equivariance, and yields heteroscedastic uncertainty via $g_\theta$. We integrate with a stochastic runge-kutta (srk) as shown in equation 11

$$z_{t_1} \;=\; \mathtt{sdeint}\big(f_\theta, g_\theta, z_0, \{t_k\}_{k=0}^{K}\big), \tag{11}$$

using a *Brownian interval* $W_t$ over $[t_0, t_K]$. The refined physical-space trajectory over the window is obtained by decoding the output of neural sde back to physical space as given in equation 12

$$\hat{x}(t_1{:}t_K) \;=\; \mathcal{D}\big(z_{t_1:t_K}, x_{\text{skip}}\big). \tag{12}$$

where $\mathcal{D}$ is a 3D decoder (transpose conv + concatenation with $x_{skip}$) which reconstructs refined fields over the window; the skip tensor $x_{\text{skip}}$ injects mid-level encoder features to preserve spatial detail and stabilize reconstruction. The complete architecture and encoder-decoder structure is shown in Figure 1.

**Ensemble refinement and uncertainty.** To quantify process uncertainty, we draw $K$ independent Brownian paths and solve the SDE for each:

$$z^{(k)} \;=\; \text{SDE\_integrate}\big(f_\theta, g_\theta, z_0, \{t_j\}; W^{(k)}\big), \qquad \hat{x}^{(k)} \;=\; \mathcal{D}\big(z^{(k)}, x_{\text{skip}}\big), \quad k = 1, \ldots, K. \tag{13}$$

Each realization uses the same initial condition $z_0$ but a different Brownian path $W^{(k)}$, yielding a sample from the predictive distribution over trajectories in the window.

We summarize the ensemble with the mean and variance:

$$\mu \;=\; \frac{1}{K} \sum_{k=1}^{K} \hat{x}^{(k)}, \qquad \Sigma \;=\; \frac{1}{K-1} \sum_{k=1}^{K} \big(\hat{x}^{(k)} - \mu\big)^2. \tag{14}$$

The mean $\mu$ serves as the refined state passed to the next auto-regressive window, while $\Sigma$ provides a time–space–channel uncertainty estimate suitable for calibration and downstream risk assessment.

### 4.5 Uncertainty Quantification

The existing climate emulators including the probabilistic ones such as spherical Dyffusion by Cachay et al. (2024) are trained with a *deterministic* objective such as MSE/RMSE, which fits the conditional mean but leaves the predictive spread under-determined. To learn both the mean and a heteroscedastic variance, we train with a latitude-weighted Gaussian negative log–likelihood (NLL) objective.

Consider a global dataset $\mathcal{D} = \{(t_i, y_i)\}_{i=1}^{N}$, where each frame $y_i \in \mathbb{R}^{C \times H \times W}$ lies on a regular lat–lon grid. At time $t_i$ the model predicts cell-wise mean $\mu_\theta(t_i)$ and variance $\sigma_\theta^2(t_i)$. The latitude-weighted Gaussian negative log-likelihood with a variance prior is given by equation 15

$$\mathcal{NLL} = -\frac{1}{NCHW} \sum_{i=1}^{N} \left[ \sum_{p=1}^{CHW} \tilde{w}_p \, \log \mathcal{N}\big(y_{i,p} \,\big|\, \mu_{\theta,i,p}, \sigma_{\theta,i,p}^2\big) \;+\; \log \mathcal{N}^+\big(\sigma_{\theta,i}^2 \,\big|\, 0, \lambda_\sigma^2 \mathbf{I}\big) \right] \tag{15}$$

where $\mathcal{D} = \{(t_i, y_i)\}_{i=1}^{N}$: dataset of $N$ time points; each frame $y_i \in \mathbb{R}^{C \times H \times W}$. Index $p = 1, \ldots, CHW$ is a flattened index over channel $c$, latitude row $\ell$, longitude column $m$; $y_{i,p}$, $\mu_{\theta,i,p}$, and $\sigma_{\theta,i,p}^2$ denote the observed value, predicted mean, and predicted variance at a single space–channel location of frame $i$; $\tilde{w}_p$ is the latitude weight for location $p$, $\log \mathcal{N}$ is Gaussian log-likelihood at location $p$; $\log \mathcal{N}^+$ is half-Gaussian prior over the non-negative variance vector of frame $i$ with scale $\lambda_\sigma$ which shrinks extreme variances and stabilizes training.

## 5 Experiments and Results

**Task:** The goal of PACER is to auto-regressively emulate state variables from initial condition and a parallel time series of ghg emissions. We train our model on 1-year and validate on 10-year roll-outs. We also validate the generalisation of our methodology using zero-shot emulation. We compare PACER against Unet and ConvLSTM baselines provided by ClimateSet. We also compare against a simple SFNO Bonev et al. (2023) given in LUCIE. The hyperparameter details for adaptation of all baselines are given in Appendix C.

**ClimateSet:** We train PACER on a total of 20 climate models, 19 provided by ClimateSet Kaltenborn et al. (2023) and FV3GFS (Zhou et al., 2019). ClimateSet compiles climate data from the Coupled Model Intercomparison Project Phase 6 (CMIP6) (Eyring et al., 2016) , incorporating climate model outputs from ScenarioMIP (O'Neill et al., 2016) and future emission trajectories of climate forcing agents from Input Datasets for Model Intercomparison Projects (Input4MIPs) (Durack et al., 2017). Each CMIP6 climate model has been standardized to a spatial resolution of 250km i.e. $96 \times 144$ grid points (latitude $\times$ longitude) with a monthly temporal resolution. Both input and output datasets consist of 86-year time-series data spanning four SSP scenarios (SSP1-2.6, SSP2-4.5, SSP3-7.0, SSP5-8.5) from 2015 to 2100. We train on SSP1-2.6, validate on SSP3-7.0, and test on SSP2-4.5 scenarios.

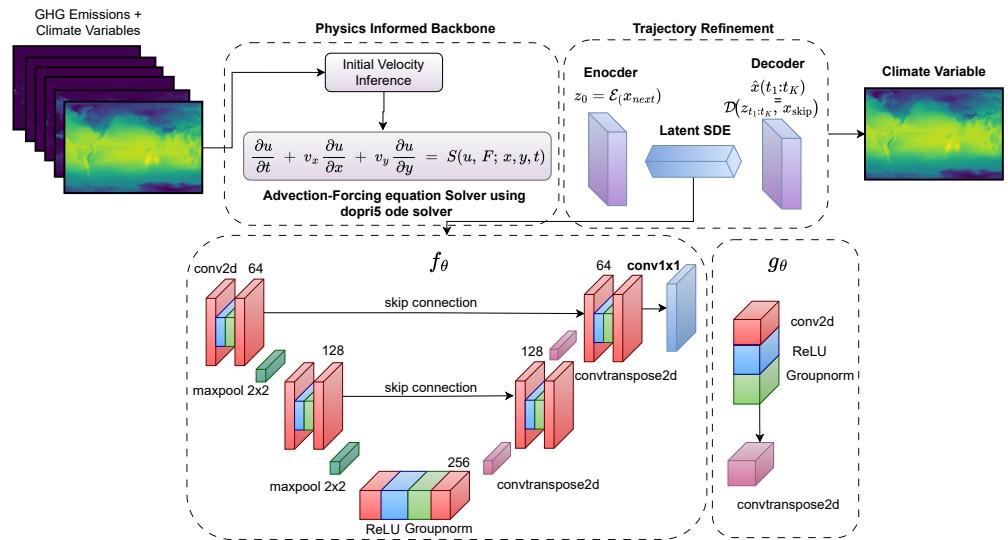

Figure 1: Complete architectural pipeline of PACER.

**FV3GFS:** We also train PACER on FV3GFS, a climate model used at the US National Weather Service and US National Centers for Environmental Prediction. It consists of 11-member initial-condition ensemble, each a 10-year integration saved every 6 hours. Forcings consist of annually repeating climatological sea-surface temperature (1982–2012 mean) and top-of-atmosphere insolation. Model output is conservatively regridded from the FV3 cubed-sphere to a $1°$ Gaussian grid $180 \times 360$ grid point and passed through a spherical-harmonic analysis–synthesis to suppress high-latitude artifacts. We train on 100 years drawn from 10 ensemble members and evaluate on a distinct 10-year member. We downsample it to a monthly cadence to align with ClimateSet baselines and ensure consistent training. The detailed list of input (diagnostic and prognostic) and output variables (diagnostic) is given in Appendix D

**Evaluation Metrics** We evaluate all models using latitude-weighted Root Mean Square Error (RMSE) for deterministic approach Nguyen et al. (2023a) and Continuous Rank Probability Square (CRPS) Winkler et al. (1996) for probabilistic approach which are described in Appendix B.1.

## 5.1 RESULTS

We report RMSE and CRPS for PACER, UNet, ConvLSTM and SFNO. The results for all Climateset's physics based numerical climate models are shown in Table 1. The results for emulating temperature at different surface levels for physics based climate model FV3GFS are given in Table 2. The best performing ML based climate emulators are emboldened. PACER outperforms all ML based models for emulating surface air temperature for auto-regressive 10 year long simulations across 20 climate models.

## 5.2 ZERO-SHOT EMULATION

We perform zero-shot emulation experiments to test the generalization capabilities of ML models on three physics based climate models: AWI-CM-1-1-MR, EC-Earth3 and TaiESM1. We report the RMSE and CRPS for TAS (surface air temperature) pre-trained on different climate models and tested on these three climate models and results are shown in Table 3. The metric for best performing model is emboldened. The pretrained climate model column shows which dataset the ML model was initially trained on before being tested on the either of the three climate models. Overall, PACER outperforms all ML models on 1-year auto-regressive roll-out for zero-shot emulation.

Table 1: 10-year roll-out results on 19 climate models which are a subset of ClimateSet. We report the RMSE and CRPS for TAS (surface air temperature).

| Climate Model | PACER | | UNet | ConvLSTM | SFNO |
|---|---|---|---|---|---|
| | RMSE | CRPS | RMSE | RMSE | RMSE |
| AWI-CM-1-1-MR | **0.369** | 0.251 | 0.916 | 0.543 | 0.406 |
| BCC-CSM2-MR | **0.381** | 0.263 | 0.950 | 0.549 | 0.412 |
| CAS-ESM2-0 | **0.420** | 0.306 | 1.431 | 0.542 | 0.451 |
| CESM2-WACCM | **0.333** | 0.233 | 1.154 | 0.556 | 0.463 |
| CESM2 | **0.353** | 0.248 | 0.926 | 0.557 | 0.475 |
| CMCC-ESM2 | **0.387** | 0.266 | 1.173 | 0.552 | 0.451 |
| CMCC-CM2-SR5 | **0.396** | 0.274 | 0.980 | 0.553 | 0.499 |
| CNRM-CM6-1-HR | **0.384** | 0.266 | 0.943 | 0.543 | 0.411 |
| EC-Earth3 | **0.372** | 0.251 | 1.284 | 0.548 | 0.383 |
| EC-Earth3-Veg | **0.399** | 0.273 | 0.987 | 0.549 | 0.418 |
| EC-Earth3-Veg-LR | **0.405** | 0.278 | 1.121 | 0.545 | 0.500 |
| FGOALS-f3-L | **0.374** | 0.266 | 1.017 | 0.533 | 0.399 |
| GFDL-ESM4 | **0.358** | 0.252 | 0.923 | 0.539 | 0.424 |
| INM-CM4-8 | **0.348** | 0.246 | 0.954 | 0.527 | 0.462 |
| INM-CM5-0 | **0.369** | 0.261 | 1.128 | 0.529 | 0.394 |
| MPI-ESM1-2-HR | **0.352** | 0.240 | 0.668 | 0.547 | 0.440 |
| MRI-ESM2-0 | **0.374** | 0.261 | 0.944 | 0.565 | 0.414 |
| NorESM2-MM | **0.363** | 0.254 | 1.044 | 0.555 | 0.420 |
| TaiESM1 | **0.353** | 0.244 | 0.989 | 0.535 | 0.398 |

Table 2: 10-year auto-regressive roll-outs on the physics based on physics based FV3GFS dataset. We evaluate near-surface air temperature (upto seven vertical levels) using RMSE and CRPS aggregated over space and time across the roll-out horizon. (Lower is better).

| Climate Variable | PACER | | UNet | ConvLSTM | SFNO |
|---|---|---|---|---|---|
| | RMSE | CRPS | RMSE | RMSE | RMSE |
| $T_1$ | **0.437** | 0.347 | 0.924 | 0.653 | 0.466 |
| $T_3$ | **0.408** | 0.274 | 0.865 | 0.656 | 0.482 |
| $T_4$ | **0.355** | 0.222 | 0.796 | 0.592 | 0.491 |
| $T_5$ | **0.349** | 0.221 | 0.899 | 0.536 | 0.379 |
| $T_6$ | **0.379** | 0.240 | 0.738 | 0.582 | 0.459 |
| $T_7$ | **0.362** | 0.237 | 0.729 | 0.562 | 0.381 |

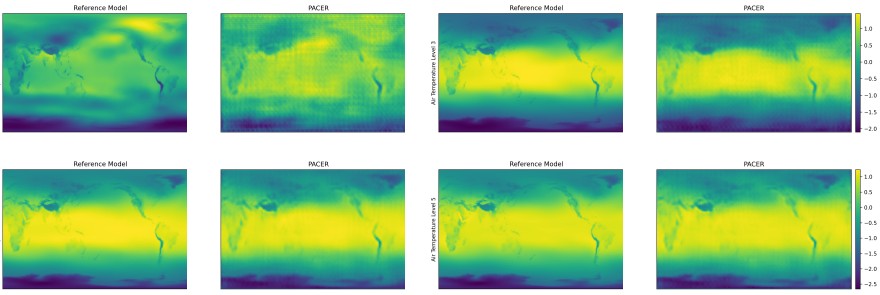

Figure 2: Time–mean maps for air temperature fields across vertical levels (2,3,4,5). Both the quantitative results in table 2 and visual results indicate stronger fidelity at higher levels (L4–L5) than at lower tropospheric levels (L2–L3), with reduced discretizations.

Table 3: Zero-shot Emulation results for 1-year roll-out on AWI-CM-1-1-MR, EC-Earth3 and TaiESM1

| Climate Model | PACER | | UNet | ConvLSTM | SFNO |
|---|---|---|---|---|---|
| | RMSE | CRPS | RMSE | RMSE | RMSE |
| AWI-CM-1-1-MR | **0.190** | 0.134 | 0.534 | 0.589 | 0.322 |
| EC-Earth3 | **0.214** | 0.149 | 0.759 | 0.590 | 0.342 |
| GFDL-ESM4 | **0.198** | 0.136 | 0.603 | 0.599 | 0.325 |
| FGOALS-f3-L | **0.225** | 0.155 | 0.659 | 0.587 | 0.351 |
| Climate Model | PACER | | UNet | ConvLSTM | SFNO |
| | RMSE | CRPS | RMSE | RMSE | RMSE |
| EC-Earth3 | **0.193** | 0.136 | 0.606 | 0.594 | 0.406 |
| TaiESM1 | **0.216** | 0.149 | 0.569 | 0.599 | 0.464 |
| CNRM-CM6-1-HR | **0.197** | 0.138 | 0.591 | 0.610 | 0.476 |
| INM-CM4-8 | **0.214** | 0.146 | 0.581 | 0.598 | 0.495 |
| Climate Model | PACER | | UNet | ConvLSTM | SFNO |
| | RMSE | CRPS | RMSE | RMSE | RMSE |
| TaiESM1 | **0.193** | 0.136 | 0.633 | 0.576 | 0.432 |
| INM-CM5-0 | **0.194** | 0.138 | 0.661 | 0.617 | 0.479 |
| MPI-ESM1-2-HR | **0.205** | 0.144 | 0.646 | 0.580 | 0.461 |
| NorESM2-MM | **0.202** | 0.140 | 0.657 | 0.579 | 0.471 |

## 5.3 ABLATION STUDIES

**Physics informed vs Physics Uninformed:** We perform an ablation study in which we remove the advection-forcing pde and directly train on the neural sde. We observe a decline in the performance of emulation and show it in figure 3. The study shows that although climate evolves under coupled, multi-physics PDEs (Navier–Stokes, thermodynamics, etc.), injecting even a minimal physics informed advection with prescribed forcings, improves skill as compared to purely data-driven baseline.

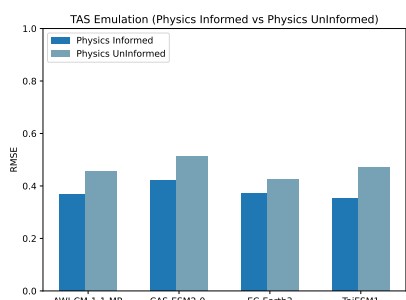

Figure 3: Physics vs. no-physics across four CMIP6 climate models

## 6 CONCLUSION AND FUTURE WORK

In this work, we present PACER, a physics and uncertainty aware climate emulator which accounts for Earth's atmospheric advection phenomenon. We incorporate a key physical law in PACER by solving a time-dependent partial different equation (PDE) using an ODE solver. Additionally, we refine the inconsistencies of the generated trajectory through a Neural SDE and account for uncertainties by explicitly training on a negative log-liklihood objective.

However, there are a few potential limitations of PACER which will be addressed as a possible future work. Our model only accounts for Advection physical law, however beyond advection, climate models solve the rotating (hydrostatic/primitive) Navier–Stokes system with moist thermodynamics, radiative transfer, turbulence/mixing and cloud microphysics, coupled to ocean circulation, land surface, and sea-ice to simulate a full multi-physics Earth system. We consider this work as a step towards building fully physics compatible climate emulators.

Additionally, PACER is trained on coarse resolution data which does not fully account for extreme events at regional level. These limitations can be addressed by training on high resolution data and encoding complex physical constraints to be able to fully emulate physics based climate models in a reliable and compute efficient way.

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

## A  IMPACT STATEMENT

Our research presents a deep learning based climate emulator which emulates temperature for multiple climate models by solving atmospheric advection. ML climate emulators run much faster and use far less energy than Global Climate Models (GCMs), making multi-year simulations with thousands of runs, quick and practical to do routinely. While ML-based climate emulators have advanced rapidly, they still lag state-of-the-art Earth System Models (ESMs) in process fidelity, long-horizon stability, and the representation of extremes. This work moves one step closer to parity by coupling a physics-aware backbone with a probabilistic correction mechanism, delivering stable 10 year roll-outs. Looking ahead, replacing full-physics solves with learned surrogates can cut runtime and energy use by orders of magnitude while preserving target statistics (means, variability, and extremes), making large ensembles, and uncertainty quantification fast and reliable, thereby accelerating climate risk assessment and reducing the carbon footprint of climate computation.

## B  PACER TRAINING DETAILS

### B.1  EVALUATION METRICS

The latitude-weighted Root Mean Square Error (RMSE) is given in equation 16 and Continuous Rank Probability Square (CRPS) for probabilistic approach is given in equation 17.

$$RMSE = \frac{1}{N} \sum_{t}^{N} \sqrt{\frac{1}{HW} \sum_{h}^{H} \sum_{w}^{W} L(i)(y_{thw} - pred_{thw})^2} \tag{16}$$

where $L(i)$ accounts for latitude weights. $L(i) = \cos(lat(i))/\frac{1}{H}\sum_{i'=1}^{H}\cos(lat(i'))$.

$$\mathrm{CRPS}\big(\mathcal{N}(\mu,\sigma^2),y\big) = \sigma\left[ z\big(2\Phi(z)-1\big) + 2\phi(z) - \frac{1}{\sqrt{\pi}} \right], \qquad z = \frac{y-\mu}{\sigma}, \tag{17}$$

## B.2 HYPERPARAMETERS

Table 4: **Left:** PACER Training Details and **Right:** PACER ConvLayer Details

| Hyperparameters | Value |
|---|---|
| Epochs | 30 |
| Conv2d kernel size | 3x3 |
| Integrator | srk |
| Noise type | diagnol |
| ODE solver | dopri5 |
| Pooling No.1 | AvgPool2d (2,2) |
| Normalization | GroupNorm |
| Activation Function | ReLU |
| Optimizer | Adam |
| Learning Rate | $1e-4$ |
| Weight Decay | $1e-6$ |
| Adam $\epsilon$ | $1e-8$ |
| LR scheduler | Exponential decay |
| Scheduler gamma | 0.98 |
| Batch size | 4 |

| ConvLayers | Value |
|---|---|
| Encoder | 4 |
| Latent SDE - Drift $f_\theta$ | 11 |
| Latent SDE - Diffusion $g_\theta$ | 1 |
| Decoder | 4 |
| Velocity Refiner | 2 |

## B.3 NUMERICAL DISCRETIZATION: IMPACT OF FINITE DIFFERENCE METHODS (FDM) ON EARTH'S SPATIAL GRIDDING IN CLIMATE MODELS

We utilize FDM for spatial discretization in PACER which divides the the physical space into a grid of discrete points. Each grid point represents a specific location, and the value of the physical quantity (e.g., temperature) is computed at each point. In FDM, continuous differential equations are approximated using discrete differences between values at specific grid points. The process of discretization converts the continuous space into a finite grid, and the differential operators (like derivatives) are approximated using differences between the values at neighboring grid points.

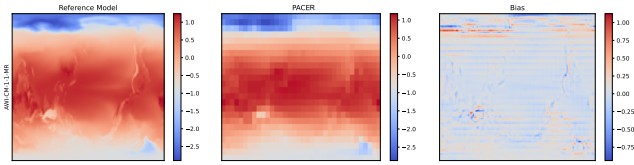

Figure 4: Numerical discretization effect on emulation of Temperature.

## B.4 HARDWARE AND SOFTWARE REQUIREMENTS

We use PyTorch Paszke et al. (2019), Pytorch Lightning Falcon (2019), torchdiffeq Chen et al. (2018) for implementation of PACER. We perform all emulator training experiments on a single NVIDIA H100_NVL GPU.

## C BASELINE HYPERPARAMETERS

We follow the same training setting for both Unet and ConvLSTM as described in ClimateSet Kaltenborn et al. (2023) except changing it from diagnostic to auto-regressive settings. We maintain the hyperparameters of the SFNO consistent with the configuration proposed in LUCIE (Guan et al., 2024).

Table 5: UNET Training Details

| Hyperparameters | Value |
| --- | --- |
| Encoder Backbone | VGG11 pre-trained on ImageNet |
| Library | segmentation-models-pytorch (SMP) |
| Encoder stride constraint | 32 (downsampling factor) |
| Input grid handling | Zero-pad lon/lat to nearest multiple of 32 |
| Output resizing | Adapted average pooling to original grid size |
| Decoder | Standard U-Net decoder (upsampling + skip connections) |
| Readout head | Linear layer |
| Temporal wrapper | Time-Distributed layer around U-Net |
| Optimizer | Adam |
| Learning Rate | $2e-4$ |
| Weight Decay | $1e-6$ |
| Adam $\epsilon$ | $1e-8$ |
| LR scheduler | Exponential decay |
| Scheduler gamma | 0.98 |

Table 6: **Left:** ConvLSTM Training Details and **Right:** SFNO Training Details

| Hyperparameters | Value | Hyperparameters | Value |
| --- | --- | --- | --- |
| Conv2d filters | 20 | SFNO blocks | 8 |
| Conv2d kernel size | 3x3 | Encoder and Decoder Layers | 1 |
| Activation Function | ReLU | Latent Dimension | 72 |
| Pooling No.1 | AvgPool2d (2,2) | Maximum Learning Rate | $1 \times 10^{-4}$ |
| Pooling No.2 | AvgPool2d (lon/2, lat/2) | Minimum Learning Rate | $1 \times 10^{-6}$ |
| LSTM Layers | 1 | Units per Layer | 32 |
| LSTM hidden units | 25 | Optimizer | Adam |
| Optimizer | Adam | Activation Function | SiLU |
| Learning Rate | $2e-4$ | Regularizer weight | $5e-2$ |
| Weight Decay | $1e-6$ | | |
| Adam $\epsilon$ | $1e-8$ | | |
| LR scheduler | Exponential decay | | |
| Scheduler gamma | 0.98 | | |

## D  DATASET VARIABLES

Table 7: ClimateSet Variables

| Symbol | Description | Usage |
| --- | --- | --- |
| TAS | Surface Air Temperature | Prognostic Variable (input & output) |
| PR | Precipitation | Prognostic Variable (input & output) |
| $CO_2$ | Carbon Dioxide | Forcing Variable (input only) |
| $CH_4$ | Methane | Forcing Variable (input only) |
| $SO_2$ | Sulfur Dioxide | Forcing Variable (input only) |
| BC | Black Carbon | Forcing Variable (input only) |

Table 8: FV3GFS Variables. The k subscript refers to a vertical layer index and ranges from 0 to 7

| Symbol | Description | Usage |
|--------|-------------|-------|
| $T_k$ | Air Temperature | Prognostic Variable (input & output) |
| DSWRFsfc | Downward shortwave radiative flux at surface | Forcing Variable (input only) |
| DLWRFsfc | Downward longwave radiative flux at surface | Forcing Variable (input only) |
| ULWRFsfc | Upward longwave radiative flux at surface | Forcing Variable (input only) |
| USWRFsfc | Upward shortwave radiative flux at surface | Forcing Variable (input only) |

