# OpenReview forum: "PACER: Physics Informed and Uncertainty Aware Climate Emulator"
_ICLR.cc/2026/Conference — ICLR 2026 Conference Withdrawn Submission_

### Official Review · Reviewer_Zqwu · 2025-10-27

**Soundness:** 1
**Presentation:** 1
**Contribution:** 2
**Rating:** 2
**Confidence:** 5

**Summary:**

The paper proposes PACER, a 2.1M parameter climate emulator designed for auto-regressive simulation. Its core contribution is a hybrid ODE-SDE framework, where a physics-based ODE models the advection-forcing dynamics , and a neural SDE refines the trajectory while capturing stochastic variability. The model predicts the mean and stddev of a normal distribution, and can thus be trained on a Gaussian NLL. The model demonstrates stable 10-year auto-regressive rollouts, at monthly resolution, and promising performance across 20 climate models.

**Strengths:**

- Physics-informed design for a climate emulator is a great direction that could benefit the emulators' trustworthiness and robustness
- Better RMSE scores than the deterministic baselines presented in the paper.

**Weaknesses:**

1. Sloppy writing hurts the flow of reading and makes the writing feel rushed. E.g., in the abstract: *"PACER is trained across is trained across"*, use proper upper-cases (ghg, unet, sde, pde...), lines 51-53 and 115-117, name SDE_integrate and sdint consistently, enocder -> encoder in Fig. 1, multiple surface levels is wrong (same issue in Table 2; only one level can be the surface level...) etc.
2. The proposed emulator is trained/tested on data at monthly resolution only. This is extremely coarse. For example, all the noted emulators in lines 96-102 run at 6 hourly resolution. The coarser resolution makes autoregressive modeling a much easier task (the model now needs to predict monthly averages instead of high-frequency weather). If the paper compared to one of these existing emulators and identified benefits for using monthly data, I'd be more inclined to let this pass.
3. Besides, PACER only emulates temperature. That's certainly a very important variable, but again, the existing emulators already go beyond that and emulate the whole atmosphere (e.g. humidity, winds, precip. etc.). Both the coarse temporal resolution and limited variable set reduce the practicability of the emulator, reduce problem complexity, and make it hard to compare it with existing approaches.
4. Relatedly, the model learns a Gaussian fit to the data. That might work for variables like temperature but certainly not for many other important variables like precip. This needs to be mentioned as a limitation. Besides, none of the results are for non-temperature variables, even though precip. is part of the used ClimateSet dataset and the original FV3GFS data (as well as many more variables which were dropped in this paper without any explanation for why).
5. Strong artefacts in the predicted global maps (e.g., see Fig. 2) raise the question of how good this emulator actually is.
6. It's unclear how the reported RMSE/CRPS are computed. Are these timestep-wise (i.e. month by month) scores, or computed on time-mean's?
7. No runtime benchmark, which is important to include for emulators.
8. The claim *"deterministic objective such as MSE/RMSE, which fits the conditional mean but leaves the predictive spread under-determined"* is inaccurate. Most diffusion models are trained on an MSE-based loss, but their posterior distribution is still quite strong.

**Questions:**

The RMSE scores in ClimateSet (fig. 6) seem to be significantly better than the ones reported here (Table 1). I understand that ClimateSet models it as a diagnostic-type task, while PACER is an autoregressive model, but are there any other contributors to this mismatch? Does this imply that diagnostic-type emulation is superior? If so, maybe the paper would benefit from a different perspective? In any case, simply omitting the existing results from climateSet without discussion is dissatisfying.

---

### Official Review · Reviewer_Dh43 · 2025-10-28

**Soundness:** 2
**Presentation:** 1
**Contribution:** 3
**Rating:** 2
**Confidence:** 4

**Summary:**

The paper proposes a climate model emulator that first resolves the advection equation, before refining the trajectory with a Neural SDE. The Neural SDE allows for accurate forecasting as well as uncertainty quantification, while incorporating the advection equation improves the skill of the model.
The model is evaluated on two datasets: ClimateSet, a dataset containing TAS, PR, and forcings data from monthly-resolution climate models, and FV3GFS, which represents 7 levels of temperatures and radiative fluxes. It outperforms competing models in TAS RMSE.

Overall, the paper shows an interesting model, with a physics-backbone, but I am not sure that the chosen physics backbone is suited for the task of climate modelling and the limited evaluation does not provide enough evidence that it is.

**Strengths:**

The idea of combining a physics-guided backbone with a stochastic residual corrector is a very compelling idea, that could be very useful in the field of climate modelling as uncertainty representation + physical consistency are two main goals. Previous research had proposed to use a physics-guided backbone with a deterministic residual corrector for weather modelling (Kochkov et al., 2023).

Resolving the advection process with FDM makes sense, and the parameterization of the Neural SDE + Uncertainty quantification is sound.

**Weaknesses:**

First, the advection diffusion model describes the evolution of gases in the atmosphere over weather timescales (i.e. high temporal resolution), but not over monthly timescale, where the dynamics of sea-surface temperature are mostly driven by the ocean. The diffusion of forcings at a monthly temporal resolution are not at all described by the advection diffusion model, but is rather dependent on the boundary conditions. Similarly, the climate-velocity index or "local migration speed" was originally defined for tracking climate change i.e. estimating the velocity between two 30-years periods (Gaponenko et al., 2022). Computing it from monthly data does not make much sense as it will be largely dependent on the initial conditions (it might depend on the season, for example).

Although the idea of using a physics-backbone for climate model emulation is compelling, I don't think that the chosen backbone (advection equation) is suited for the task. Such a model (on monthly data) would probably be better suited for modelling ocean dynamics.

The authors empirically show that the RMSE is lower when using the physics-backbone. However, the evaluation is very limited and this is not enough for proving that the advection equation truly represents the climate dynamics.

First, the result in figure 3 does not show statistical significance. I recommend training and evaluating the models with multiple seeds + initial conditions to get error bars and statistical significance. This could also be used to evaluate CRPS for competing models (as is traditionnally done with climate ensembles).

Second, when using the advection equation, the model is informed of the "initial velocity", which might lead to better MSE in the next months prediction only, and could lead to the results shown in figure 3. Using metrics that reflect long-term vs short-term prediction is necessary here.

Third, RMSE is not by itself enough to verify whether a climate emulator performs well. It favors smooth models (double-penalty of MSE) but more importantly, the goal of climate modelling is to capture climate trends, as the exact temperature on a specific month is not predictable. Thus, I would recommend reporting statistics like the mean + std. dev., or the power spectral density of the climate indices as done in [1], or the time-dependent area-weighted global mean and the area-weighted global mean bias and RMSE of time-mean fields as done in [2].

Fourth, the models are not evaluated on precipitation, although there are only two output variables in climateset, temperature and precipitation. No performance on precipitation is reported, and the second dataset contains temperature at different levels. I understand that the goal is to emulate TAS, as a response to forcings, but emulating the interactions between variables is key for this task, and understanding the model's performance on precipitation, and whether or not the advection equation helps is very important.

The model claims to perform uncertainty quantification. CRPS is reported but not compared. Also, no maps of the learned uncertainty is shown. It is also not clear if the uncertainty is well propagated during the autoregressive process. As mentioned above, baselines could be trained with multiple seeds and initial conditions to compute CRPS.

The limited evaluation prevents from understanding the importance of the two model components, whether PACER actually outperforms baselines, and whether the choice of advection equation as a physics-backbone is supported by empirical evidence.

Finally, the presentation of the paper could be improved. The zero-shot emulation section is hard to grasp ("We report the RMSE and CRPS for TAS (surface air temperature) pre-trained on different climate models"; no "pretrained climate model" column in table 3). Figure 2 illustrates "stronger fidelity", but does not show any comparison, and labels are small and blurry. In section 4 and Figure 1, it is not clear what u or x are (x,y are coordinates in the left part, but x is then the variable of interest in the right part of Figure 1?)

[1] Hickman et al., Causal Climate Emulation with Bayesian Filtering, 2025

[2] Watt-Meyer et al., ACE: A fast, skillful learned global atmospheric model for climate prediction, 2023

**Questions:**

Was there any hyperparameter search that was done when running baselines on the FV3GFS dataset? Or did you reuse the parameters from ClimateSet?

---

### Official Review · Reviewer_Ab3o · 2025-10-31

**Soundness:** 1
**Presentation:** 1
**Contribution:** 2
**Rating:** 2
**Confidence:** 4

**Summary:**

This paper proposes PACER, a 2.1M parameter climate model emulator that combines a physics-informed ODE backbone with a neural SDE refinement module. The model is trained to predict surface temperature and precipitation fields from atmospheric GHG concentrations, using data from ClimateSet and FV3GFS. The authors demonstrate stable 10-year autoregressive rollouts and zero-shot generalization capabilities, outperforming selected UNet, ConvLSTM, and SFNO baselines on deterministic and probabilistic metrics.

**Strengths:**

1. **Novel architecture**: The ODE-SDE framework is interesting, explicitly incorporating advection dynamics while using neural SDEs for uncertainty quantification.
2. **Multi-model training**: Evaluation across 20 climate models demonstrates broader applicability than an emulator trained on a single model.
3. **Uncertainty quantification**: Training directly on NLL with multiple stochastic trajectories provides principled heteroscedastic uncertainty estimates.
4. **Parameter efficiency**: At 2.1M parameters, PACER is substantially smaller than state of the art emulators such as ACE. (However, I'm not convinced that these other models are prohibitively large.)
5. **Zero-shot generalization**: Table 3 demonstrates some cross-model transfer capability.

**Weaknesses:**

1. **Choice of physics prior**: Surface air temperature and precipitation are not advective processes. Indeed, in physics-based climate models, they're not prognostic variables at all, but rather are diagnosed. An advection equation therefore seems like a poorly motivated physics prior for variables that are not governed by a PDE.
2. **No computational timing results**: The paper's core motivation, and a primary motivation for climate model emulation generally, is computational efficiency. However, the paper provides no wall-clock time comparisons. Solving both an ODE and an SDE at each autoregressive step could be expensive. Without timing data, efficiency claims are unsupported.
3. **Physics consistency**: The authors claim that PACER is "physically consistent," but this claim is not supported. Beyond RMSE/CRPS, one might typically look conservation of mass/momentum/energy, but that isn't possible here (at least not for the ClimateSet dataset consisting of only surface temperature and precipitation). Other options would be: spatial power spectra, temporal autocorrelation functions, climate statistics.
4. **GHG forcing**: The $S(u, F; x, y, t)$ term in Eq. 3 is never actually specified. How are "the GHG forces" incorporated into the PDE? It's not obvious to me how this forcing would look.
5. **Unclear boundary condition handling**: No discussion of spherical geometry, for example, periodic boundaries and singularities at poles.
6. **Confidence intervals**: Tables present single numbers with no error bars, confidence intervals, or significance tests.
7. **Visual artifacts**: Figure 2 shows severe artifacts in PACER predictions that aren't addressed by the authors.
8. **Ensemble setup**: The ensemble mean at a given window serves as the input to the next window. The SDE then provides a measure of the spread within that window. However, this also means that the "ensemble" is not continuous across windows, that is to say, ensemble member $i$ in window $w$ does not follow from ensemble member $i$ in window $w-1$. I think this distinction, that PACER does not truly simulate an ensemble during autoregressive rollout, should be made very clear in the text.
9. **Velocity inference methodology (Section 4.3)**: This section is convoluted and poorly explained. The authors claim that "The standard approach is the climate-velocity index." I disagree that this approach is "standard" for approximating derivatives in general circulation models. Indeed, the cited paper (Gaponenko et al., 2022) aims to solve a different problem, that is, estimating the velocity of isotherms under climate change, which is distinct from the PDE being modeled in this paper. Also, the climate velocity should be $$v = \frac{1}{\|\nabla u\|}\frac{\partial u}{\partial t},$$ not $v =\|\nabla u\|\frac{\partial u}{\partial t}$ as it's written in the paper. This makes the units incorrect. Which version did you implement in practice?

**Questions:**

1. Can you clarify the velocity inference method outlined in Section 4.3? Why is this necessary at all? What exactly is $v$ in Eq. 5? Did you use the equation exactly as it appears in Eq. 5, with the wrong units? If so, how does this affect your results?
2. Why is an advection-based prior an appropriate choice for TAS and precipitation?
3. Can you provide wall-clock timing comparisons against GCMs and baseline emulators?
4. How exactly is $S(u, F; x, y, t)$ parameterized? How do GHG forcings enter the PDE?
5. How are spherical geometry boundary conditions handled?
6. Can you provide physics-based validation metrics (spectra, etc.)?

---

### Official Review · Reviewer_3eAm · 2025-11-01

**Soundness:** 2
**Presentation:** 1
**Contribution:** 2
**Rating:** 2
**Confidence:** 4

**Summary:**

PACER is an uncertainty-aware climate emulator that integrates a physics-based advection PDE with a Neural SDE residual corrector. The proposed method enables stable 10-year temperature field emulation on multiple climate models.

**Strengths:**

1. The paper proposes a novel autoregressive framework that combines an advection PDE-based physics-informed backbone with a Neural SDE-based uncertainty-aware residual corrector.
2. Training the neural SDE under a Gaussian NLL objective with multiple Brownian realizations ($K>1$) enables heteroscedastic aleatoric uncertainty estimation and reduces gradient variance.
3. The explicit integration of advection dynamics via ODE solver (dopri5) with velocity field inference provides a transparent physics backbone that is computationally lightweight.

**Weaknesses:**

1. Despite claiming to be "Physics Informed," the physical laws integrated into the emulator are limited only to advection, which is far from the complex multi-physics systems of actual GCMs.
2. PACER emphasizes being "lightweight" (2.1M parameters) compared to ACE (200M parameters) and Spherical Dyffusion (200M parameters), but in the results section, it only compares performance with UNet, ConvLSTM, and SFNO instead of these models.
3. There is only one ablation (Figure 3) that compares "physics informed vs uninformed" but doesn't isolate individual components.
4. The notation lacks consistency, reducing readability. Eq.2 uses $x_t$, while subsequent sections use $u_t$ for the climate state. Forcing is denoted as $F$ in some places and $f$ in others.
5. The authors claim their proposed method is "lightweight," but there is no wall-clock time comparison, GPU memory usage, or FLOP analysis.

**Questions:**

1. Can you provide quantitative evidence that learned velocity fields (after refinement) maintain physical plausibility? For example, the autghors can show visualizations compared to reanalysis winds.
2. Can you compare performance using Euler and RK4 solvers instead of the dopri5 solver? Can you compare RMSE and wall-clock time for different ODE solvers?
3. Do PACER and the baselines have the same number of encoder and decoder layers? What are the parameter counts for PACER and each baseline? Could you provide comparisons under a reasonably fair parameter budget?
4. Why was no direct comparison performed with studies [1,2] that similarly integrated the Advection-Diffusion Equation into a Neural Network framework? Can you discuss further what differentiates your work from these related studies?


> [1] "Climate modeling with neural advection–diffusion equation." Knowledge and Information Systems 65.6 (2023): 2403-2427.
>
> [2] “Climode: Climate and weather forecasting with physics-informed neural odes." ICLR 2024

---

### Official Review · Reviewer_vkm3 · 2025-11-06

**Soundness:** 2
**Presentation:** 1
**Contribution:** 2
**Rating:** 2
**Confidence:** 3

**Summary:**

The paper proposes physics-aware architecture for climate. The method is based on advection, and incorporates a neural SDE.

**Strengths:**

The method beats the baselines, and is able to learn climate.

**Weaknesses:**

The clarity of the paper is subpar. There are typos, the math is inconsistent and the method pipeline is difficult to follow. The model is given as a long storyline describing different parts of the system, without giving a clear beginning or overall framework. I'm quite confused how the SDE, the ODE and the latent modelling plays together, or what is being learnt.

The novelty is low: the paper combines relatively straightforward PDE equations with a standard neural SDE.

The baselines are overly simple. Comparing to just Unet or LSTM is ok as naive baselines, but they are not serious competing methods. The SFNO is a good baseline, but there is a total lack of any bespoke climate or physics models for this problem. The results lack standard deviations.

**Questions:**

See above.

---

### Note · Authors · 2025-12-03

I have read and agree with the venue's withdrawal policy on behalf of myself and my co-authors.